# Learning label smoothing for text classification

Han Ren[1,2], Yajie Zhao[3], Yong Zhang[4] and Wei Sun[5]

[1] Laboratory of Language Engineering and Computing, Guangdong University of Foreign Studies, Guangzhou, China
[2] Laboratory of Language and Artificial Intelligence, Guangdong University of Foreign Studies, Guangzhou, China
[3] School of Information Science and Technology, Guangdong University of Foreign Studies, Guangzhou, China
[4] School of Computer Science, Central China Normal University, Wuhan, China
[5] School of Information Science and Technology, Qiong Tai Normal University, Haikou, China

## ABSTRACT

Training with soft labels instead of hard labels can effectively improve the robustness and generalization of deep learning models. Label smoothing often provides uniformly distributed soft labels during the training process, whereas it does not take the semantic difference of labels into account. This article introduces discrimination-aware label smoothing, an adaptive label smoothing approach that learns appropriate distributions of labels for iterative optimization objectives. In this approach, positive and negative samples are employed to provide experience from both sides, and the performances of regularization and model calibration are improved through an iterative learning method. Experiments on five text classification datasets demonstrate the effectiveness of the proposed method.

## INTRODUCTION

The benchmark performances of natural language processing applications are constantly pushed by the increasing model complexity in the past decades (*Chen et al., 2021*). Complex models contain complicated encoding and decoding structures as well as significant numbers of parameters, which may lead to model overfitting (*Zhang et al., 2018*), which means that a model performing well in the training stage achieves low performance in the testing stage. The main reason for this is insufficient training data and noise interference (*Ying, 2019*). To address this problem, a wide range of regularization techniques have been investigated, considering both generalization and training errors (*Srivastava et al., 2014*).

Label smoothing (LS) (*Szegedy et al., 2016*) is a type of label regularization that provides more reasonable class labels. The basic idea of LS is to change the optimization objective from one-hot target to a value between 0 and 1. It normally adds noises to the model, to mitigate the problem of overfitting. This idea encourages the model to learn generalizable representations and make calibrated predictions. LS is also widely used in text classification models (*Desai & Durrett, 2020*; *Liu et al., 2022*).

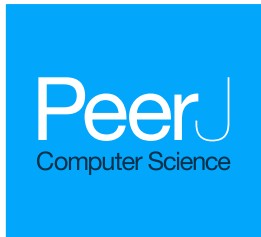

Corresponding author
Wei Sun, sun@mail.qtnu.edu.cn

However, many studies on LS add uniform noise to the models, neglecting the relationships between categories. For instance, when the target is *tea*, it would be inappropriate to apply the same degree of smoothness to *coffee* and *CPU*. The predicted probability of *coffee* increases while the probability of *CPU* decreases. To create more reasonable labels, several studies have been conducted to improve LS. In dialog generation, *Wang et al. (2021b)* used an auxiliary distribution and one-hot distribution weighting. *Saha, Das & Srihari (2022)* transformed a uniform distribution into a more natural distribution based on semantics. In image classification, *Maher & Kull (2021)* investigated the utilization of a teacher network to guide non-target probabilities. Among studies conducted on node classification, *Zhou et al. (2023)* and *Wang et al. (2021a)* represented labels as graphs, propagating node information to aggregate neighboring distributions to determine an appropriate node representation. In text classification, *Luo, Xi & Mao (2021)* proposed a label smoothing method using a fake label, but failed to explain the role played by the fake label. Margin-based label smoothing (*Liu et al., 2022*) imposes a controllable margin on logit distances, penalizing the distances exceeding a specified margin. Although the above methods can enhance model generalization, they do not consider the impact of incorrect examples on the model.

In this study, we propose an adaptive label smoothing method to address the problem of non-target distribution by learning soft label distributions during the training process. We argue that the probabilities of non-target classes should be positively correlated with similar ground-truth labels; that is, the greater the similarity to the real labels, the higher the probability. Inspired by the work of *Ding et al. (2019)*, we developed an adaptive label regularization method to adjust the strength of regularization, benefiting from erroneous experiences. For classes in which instances are often misclassified, stricter constraints should be adopted to improve model performance, considering that the model may not be learning adequate information or may even be underfitting. In summary, the differences between the proposed method and current approaches to LS are two folds: 1) few studies of LS have discussed the restricted generalization imposed by incorrect cases, whereas this study considers erroneous examples; 2) in contrast to other models, our approach explicitly considers the impact of excessive regularization, aiming to strike a balance between regularization and performance.

The major contributions of this study are as follows:

• A novel method, discrimination-aware label smoothing (DALS), is proposed based on negative samples to alleviate the underfitting problem caused by excessive regularization.

• The model learns and obtains adaptive soft labels through a training process requiring neither external knowledge nor changes to the original structure of the model. Thus, it is applicable to any backbone model.

• Experiments on several benchmark datasets indicate that the proposed method addresses the problem of overfitting and achieves competitive improvement. The average increases in accuracy for the Ohsumed, 20NG, and R52 datasets were 5%, 2%, and 2%, respectively.

The remainder of this article is organized as follows: "Related Work" summarizes the regularization tools for labels. The calculations used for the proposed approach are

described in "Model". In "Experimental Analysis", extensive experiments on comparative analyses are presented. Finally, conclusions are drawn in "Results".

## RELATED WORK

### Label smoothing

As previously discussed, LS has boosted the performance of computer vision (*Xu et al., 2020*) and natural language processing tasks (*Lukasik et al., 2020*). Unlike LS, the unigram label smoothing developed by *Pereyra et al. (2017)* assigns the frequency of each label as the prior distribution rather than the uniform distribution. Both share a fixed prior-label distribution, which may not be satisfied by numerous complex tasks. To fill this gap, substantial advancements have been made in adaptive LS, which can be divided into two categories.

a) *Revising the uniform distribution of LS*. Incorporating the idea of the k-nearest neighbor algorithm, *Bahri & Jiang (2021)* assigned weights between the uniform distribution and the number of correct samples within radius $K$ divided by the total amount of samples. *Penha & Hauff (2021)* replaced non-target labels with negative sampler scores. *Song et al. (2020)* selected candidate words that shared the history of the previous step, thereby redefining and calculating the probability distribution of candidate words as a smoothing distribution based on context. However, these methods either require a specific model structure or are only performed for specific tasks, posing challenges for text classification tasks. In contrast, our method can be applied to any model. For image processing, *Zhang et al. (2021)* improved the loss function by accumulating the distributions of correctly classified labels to enhance image recognition; however, this method ignores the adjusted effect of negative samples on the models.

b) *Changing the smoothing factor*. *Krothapalli & Abbott (2020)* chopped images by considering the relative sizes of the objects in the training set. *Li, Dasarathy & Berisha (2020)* performed clustering on the training data and learned the smoothing intensity of each cluster. *Wei et al. (2022a)* proposed the use of a negative smoothing factor in high-noise regimes.

Our method falls into the first category, as we determine a more natural label distribution in the training process.

### Calibration

Calibration predicts the probability or confidence in the model to approximate its true accuracy. The calibrated probability is important for interpreting the model (*Guo et al., 2017*) because it reflects the confidence level in an actual scenario. Efforts aimed at estimating calibration in well-trained models are mainly divided into two classes: post-processing and model calibration. Some classic binary models that use post-processing steps include Platt scaling (*Platt, 1999*), histogram binning (*Zadrozny & Elkan, 2001*), and isotonic regression (*Zadrozny & Elkan, 2002*). For multiclass settings, temperature scaling is a competitive calibration method (*Guo et al., 2017*; *Balanya, Maroñas & Ramos, 2022*; *Khan, Wang & Liu, 2023*) prevalent in knowledge distillation (*Hinton, Vinyals & Dean, 2015*). LogitNorm (*Wei et al., 2022b*) optimizes the logit vector as a unit vector with a

constant magnitude. Model calibration introduces calibration terms for loss (*Kumar, Sarawagi & Jain, 2018*; *Mukhoti et al., 2020*), LS (*Szegedy et al., 2016*; *Wang et al., 2021b*), and data augmentation (*Thulasidasan et al., 2019*; *Yun et al., 2019*). *Pereyra et al. (2017)* were the first to propose LS for model calibration. *Müller, Kornblith & Hinton (2019)* conducted an in-depth study on LS calibration. The principle of LS is to increase the entropy of the output probability distribution to alleviate the problem of overconfidence. We also investigated the calibration effects of the proposed method.

## Label regularization

Label-correction techniques that consider label quality have been developed to prevent mistakes in handcrafted labeling. Bootstrapping loss was proposed by *Reed et al. (2014)*, which involves weighting the real labels with a predicted probability to reduce the influence of noise on parameter updating. Another approach described by *Ma et al. (2018)* decreases the weight of the hard labels over time. *Arazo et al. (2019)* integrated the concept of bootstrapping loss with dynamic weight adjustment, updating the loss of normal and noisy samples in opposite directions. Other regularization methods are employed at the loss level. For instance, *Patrini et al. (2017)* introduced a matrix $T$ to estimate the transition probability from real to noisy labels, proposing forward and backward losses based on $T$ to optimize real labels. In DisturbLabel (*Xie et al., 2016*), a few samples are randomly selected and trained using incorrect labels during each iteration. Similarly, the proposed method utilizes loss-function augmentation, which enables more flexible operations to adaptively adjust to the target distribution.

## MODEL

### Label smoothing

Let $D = \{(x_i, y_i)\}_{i=1}^{N}$, where $x_i$ denotes the $i$-th document; $Y = \{y_i \in \{0, 1\}^K\}$, where $K$ is the number of document category. When $x_i$ is fed into the deep neural network, the model outputs a K-dimensional representation. The softmax function is used in the output layer of the neural network models to predict probability $p(k|x_i)$ for class $k$. The output distribution of the model is denoted by $p$. The standard cross-entropy (CE) loss function can then be written as

$$\mathcal{L}_{hard} = H(q, p) = -\sum_{k=1}^{K} q(k|x_i) \log(p(k|x_i)) \tag{1}$$

where $q$ is the ground-truth label, which is typically a one-hot distribution; $q(k|x_i)$ is marked as 1 if and only if category $k$ is the target class and 0 otherwise. Following this, we use the backbone to denote the model with CE. However, LS does not use a one-hot distribution to calculate the loss, introducing the noise distribution $u(k|x_i)$ instead. Thus, the ground-truth label becomes

$$q'(k|x_i) = (1 - \varepsilon)q(k|x_i) + \varepsilon u(k|x_i) \tag{2}$$

and the loss is changed to

$$\mathcal{L}' = (1 - \varepsilon)H(q,\ p) + \varepsilon H(u,\ p) \tag{3}$$

where $\varepsilon$ is the smoothing factor. The loss function comprises two parts: 1) CE between the one-hot distribution and the predicted distribution $H(q,\ p)$ and 2) CE between the noise distribution and the predicted distribution $H(u,\ p)$.

During the training process, if a machine learning model becomes overconfident in its predictions, $H(q,\ p)$ approaches 0, whereas $H(u,\ p)$ increases significantly. This implies that LS introduces a regularizing effect, $H(u,\ p)$, to prevent overconfident model predictions.

In traditional LS, $u(k|x)$ follows a uniform distribution, that is, $u = \frac{1}{K}$. The loss function is expressed as follows:

$$\mathcal{L}_{ls} = -\sum_{k=1}^{K}\left[(1 - \varepsilon)q(k|x_i) + \frac{\varepsilon}{K}\right] \cdot \log(p(k|x_i)) \tag{4}$$

where $\varepsilon$ is usually set to 0.1 in LS. When $\varepsilon = 0$, this is equivalent to calculating CE using hard labels.

However, $u$ is independent of the data: $u(k|x) = u(k)$. Hence, the uniform distribution is questioned when applying the same probability distribution to incorrect labels. We assume that the label distribution correlates with the similarity between categories. One way to reduce the loss and optimize model performance is to reduce $H(u,\ p)$, specifically by making the $u$ distribution as close as possible to the predicted distribution. We posit that an iterative approach for updating soft labels is more reasonable than using fixed values, as inspired by *Zhang et al. (2021)* and *Zhou et al. (2023)*. Therefore, we designed a DALS method based on this strategy. DALS considers the real relationships between different categories and uses predictions to extract inter-class relationships that are more discriminative for the model.

## Discrimination-aware label smoothing

DALS uses category correlation in model prediction and dynamically updates soft labels during the training stage. The soft-label distribution is $u(k|x)$, which differs for each epoch. The model is supervised by the soft label calculated in the previous epoch, and the soft label is updated at the end of the current epoch. For a one-hot distribution, the probabilities for all classes are 0 except for the target class, which is marked with a probability of 1. Traditional LS employs a uniform distribution for non-target classes and reduces the probability of the target class to slightly below 1. In contrast, DALS discards the uniform distribution for the assignment of non-target classes and adaptively adjusts the label distribution.

We define $q_{x_i,k}^t$ as the soft distribution of class $k$ in the $t$-th epoch, where these distributions are specified for $x_i$. The soft label $q_{x_i,k}^{t-1}$ calculated in the $(t-1)$-th epoch will be used to guide the training process of the $t$-th epoch. The training loss at this time can be represented as

$$\mathcal{L}_{soft}^t = -\sum_{k=1}^{K} q_{x_i,\ k}^{t-1} \cdot \log(p(k|x_i)) \tag{5}$$

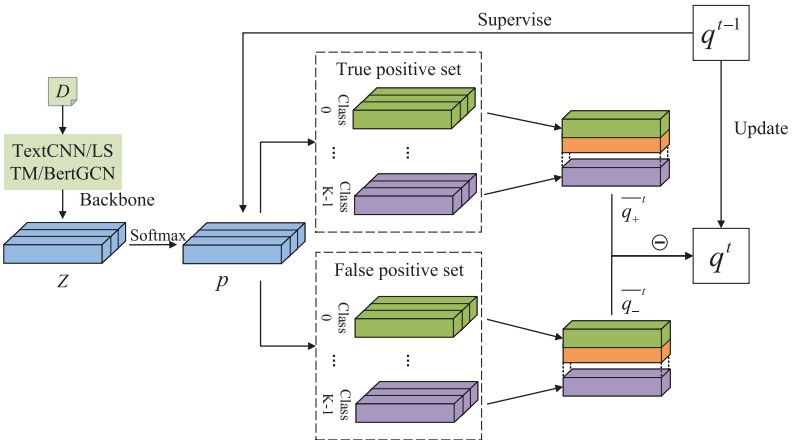

**Figure 1 The overall structure of DALS.** The overall structure of our DALS. This figure depicts the process to produce $q^t$ in the epoch of $t$. Predicted score is supervised by $q^{t-1}$ and used to calculate the loss.

Figure 1 illustrates the overall framework. Text $x_i$ can be classified using any classification backbone, such as TextCNN (*Kim, 2014*) or BertGCN (*Lin et al., 2021*). The logits from the last layer are denoted as $Z$. The predicted score $p(x_i)$ is then obtained using a softmax layer. The $p(x_i)$ scores of the true positive and false positive samples are accumulated separately.

Specifically, we denote the set of samples with the prediction class $c = \underset{k}{\mathrm{argmax}}\, p(k|x_i)$ as $D$. We also define a true positive set called $D_+$ and a false positive set called $D_-$. At the end of each epoch, the accumulated class distribution is processed to balance the score contributions of each sample. The following equations are used for the calculation:

$$
\begin{cases}
\overline{q+}^t_{x_i,k} = \dfrac{1}{|D|} \displaystyle\sum_{x_i \in D_+} p(k|x_i) \\[3mm]
\overline{q-}^t_{x_i,k} = \dfrac{1}{|D|} \displaystyle\sum_{x_i \in D_-} p(k|x_i)
\end{cases}
\tag{6}
$$

where $\overline{q+}^t_{x_i,k}$ is the soft label of class $k$ calculated using true positive samples corresponding to sample $x_i$. Similarly, $\overline{q-}^t_{x_i,k}$ is computed using false positive samples. Our soft label $q$ is calculated in two parts: $\overline{q+}$ and $\overline{q-}$. A truncation value of 0 is set to limit the impact of false positive distributions on the overall values. We define

$$
q^t_{x_i,k} = \max\left(0,\ \lambda\overline{q+}^t_{x_i,k} - \overline{q-}^t_{x_i,k}\right)
\tag{7}
$$

where $\lambda$ is a hyperparameter to balance the impact of true positive and false positive cases.

By adding hard label supervision, the updated training loss is changed to

$$
\mathcal{L}_{all} = (1-\alpha) \cdot \mathcal{L}^t_{soft} + \alpha \cdot \mathcal{L}_{hard}
\tag{8}
$$

where $\alpha$ determines the trade-off between soft and hard losses, and the value of

**Table 1 Datasets in the experiment.**

| Dataset | #Documents | #Training | #Test | #Classes | #Words |
|---|---|---|---|---|---|
| 20NG | 18,846 | 11,314 | 7,532 | 20 | 42,757 |
| R8 | 7,674 | 5,485 | 2,189 | 8 | 7,688 |
| R52 | 9,100 | 6,532 | 2,568 | 52 | 8,892 |
| Ohsumed | 7,400 | 3,357 | 4,043 | 23 | 14,175 |
| MR | 10,662 | 7,108 | 3,554 | 2 | 18,764 |

$\alpha$ represents the confidence in the hard label. When $\alpha$ equals 1, it is equivalent to calculating with the hard label. We define $\overline{q_+}^0$ as a uniform distribution, and $\overline{q_-}^0 = \mathbf{0}$ because the soft label in the 0-th epoch is unavailable, according to Eq. (5). Thus, $\hat{q}^0 \triangleq \frac{1}{K}I$, where $I$ denotes the identity matrix. In the early stage; this model is equivalent to using traditional LS.

True positive samples enable the model to generalize, allowing it to identify documents that were previously misclassified because the correct class was similar to the other classes. However, for categories with low precision, the model carries the risk of underfitting and an increased error rate. Thus, more generalization capabilities are not urgently needed. In particular, for these classes, we need to reduce the impact of true positive sample-based generalization on the model and implement a more rigorous loss assessment. Subtracting the value of $\overline{q}_{-x_i, k}^t$ weakens the regularization degree and reduces the blurring degree of the boundary between classes, making the class boundary clearer.

Thus, we propose DALS to reduce the underfitting caused by excessive regularization.

## EXPERIMENTAL ANALYSIS

### Datasets

The datasets included 20-Newsgroups (20NG), R8 and R52 in Reuters 21,578, Ohsumed, and a movie review (MR) (Table 1).

20NG has 18,846 news documents, of which 113,134 and 7,532 were used for the training and test sets, respectively, which were classified with 20 labels.

R8 and R52, extracted from Reuters 21,578, have eight and 52 categories, respectively. R8 was divided into 5,485 documents for training and 2,189 documents for testing, whereas R52 was split into 6,532 training documents and 2,568 testing documents.

The Ohsumed *Corpus* comes from the MEDLINE database, which contains bibliographies of medical literature and has been processed to retain only 7,400 documents belonging to a single category. There were 3,357 documents in the training set and 4,043 documents in the test set, which were divided into 23 classes.

MR (*Pang & Lee, 2005*) is a short-text dataset of film reviews containing one sentence for each document and is mainly used for dichotomous emotional classification. There were 5,331 positive and 5,331 negative comments.

## Baselines

The various models chosen for the baselines are listed as follows:

TextCNN (*Kim, 2014*) automatically combines and filters n-gram features to obtain high-level semantic information.

LSTM (*Hochreiter & Schmidhuber, 1997*) is a special form of recurrent neural network. The hidden state in the final step is used to represent the entire text.

FastText (*Joulin et al., 2017*), wherein the word vector and average n-gram vector are regarded as the document embedding.

TextGCN (*Yao, Mao & Luo, 2019*) constructs the entire *corpus* as a heterogeneous word-document graph, whereby the document classification problem is transformed into node classification.

SGC (*Wu et al., 2019*) reduces complexity by removing nonlinearities between the GCN layers, thereby collapsing the function into a linear transformation.

TensorGCN (*Liu et al., 2020*) constructs a text-graph tensor to describe semantic, syntactic, and sequential contextual information. Intragraph and intergraph propagations were conducted.

BERT (*Kenton & Toutanova, 2019*) and its variant RoBERTa (*Liu et al., 2019*): BERT refers to the bidirectional encoder representations from transformers that create numerous state-of-the-art models. RoBERTa is a robust, optimized BERT pre-training method.

BertGCN also builds a heterogeneous graph in which the document nodes are initialized with a pre-trained Bert. Subsequently, they are jointly trained with Bert and GCN for text classification. RoBERTaGCN, BertGAT, and RoBERTaGAT share this concept.

## Experimental setup

Five models were selected for topic classification and sentiment analysis: TextCNN, LSTM, FastText, TextGCN, and BertGCN. For TextCNN, three types of kernels with sizes of two, three, and four were set, and the number of kernels for each type was 100. For LSTM, we chose a hidden layer size of 64. In BertGCN, the [CLS] token of the output feature was treated as the document embedding. The Bert-base-uncased model from HuggingFace (https://huggingface.co/bert-base-uncased) was used following (*Lin et al., 2021*), randomly dividing 10% of the training data for validation. All models used the Adam (*Kingma & Ba, 2015*) optimizer and adopted 300-dimensional GloVe word embeddings (*Pennington, Socher & Manning, 2014*). The main parameters included the number of epochs, batch size, learning rate, early stopping, $\alpha$, and $\lambda$. Early stopping indicates that the training process is terminated in advance if the performance of the validation set does not improve within a certain number of steps. Table 2 lists the parameter configurations of different models selected for comparison. We retained the default parameters in the original methods, setting $\lambda = 1.4$ and $\alpha = 0.96$ as moderate choices for the experiments. The performance was enhanced through further tuning. The models were trained using an NVIDIA A100 Tensor Core GPU.

**Table 2  Experimental setting.**

| Parameters | TextCNN | LSTM | FastText | TextGCN | BertGCN | SGC | TensorGCN | BERT |
|---|---|---|---|---|---|---|---|---|
| Epoch | 100 | 100 | 100 | 200 | 60 | 3 | 1,000 | 60 |
| Batch size | 64 | 64 | 64 | – | 16 | – | – | 64 |
| Learning rate | 0.008 | 0.008 | 0.008 | 0.02 | 0.001 | 0.2 | 0.002 | 0.001 |
| Early stopping | 50 | 50 | 50 | 10 | – | – | 10 | – |
| Optimizer | Adam | Adam | Adam | Adam | Adam | L-BFGS | Adam | Adam |

**Table 3  Performance on test data.**

| Models | 20NG | R8 | R52 | Ohsumed | MR |
|---|---|---|---|---|---|
| TextCNN | 0.8215 | 0.9571 | 0.8759 | 0.5844 | 0.7775 |
| LSTM | 0.7543 | 0.9609 | 0.9048 | 0.5110 | 0.7733 |
| FastText | 0.7938 | 0.9613 | 0.9281 | 0.5770 | 0.7514 |
| TextGCN | 0.8634 | 0.9707 | 0.9356 | 0.6836 | 0.7674 |
| SGC | 0.885 | 0.972 | 0.940 | 0.685 | 0.759 |
| TensorGCN | 0.8794 | 0.9804 | 0.9505 | 0.7011 | 0.7791 |
| BERT | 0.853 | 0.978 | 0.964 | 0.705 | 0.857 |
| RoBERTa | 0.838 | 0.978 | 0.962 | 0.707 | 0.894 |
| RoBERTaGCN | 0.895 | 0.982 | 0.961 | 0.728 | **0.897** |
| BertGAT | 0.874 | 0.978 | 0.965 | 0.712 | 0.865 |
| RoBERTaGAT | 0.865 | 0.980 | 0.961 | 0.712 | 0.892 |
| BertGCN | 0.893 | 0.981 | 0.966 | 0.728 | 0.860 |
| BertGCN w/DALS | **0.8947** | **0.9828** | **0.9667** | **0.7361** | 0.8646 |

**Note:**
Each bold entry denotes the best performance of the metric in the column.

# RESULTS

## Performance of text classification

Experiments were conducted on five benchmark datasets, and the results are listed in
Table 3. The experimental results on the original benchmark models were obtained from
TextGCN (*Yao, Mao & Luo, 2019*) and other original studies (*Liu et al., 2020*; *Lin et al.,
2021*). The results in Table 3 show that DALS performed well on several datasets when
using the BertGCN backbone, achieving higher classification accuracy than some
traditional and outstanding methods.

Table 4 lists the accuracy and Macro-F1 results of five models with DALS. It shows that
systems with DALS outperform those without DALS in all datasets. Compared with the
baselines, the accuracy of models using DALS on Ohsumed improved by 5% on average
and by 2% on the 20NG and R52 datasets. The existence of several categories in these
datasets, where some categories are difficult to distinguish, may be a possible explanation
for this phenomenon. The enhancements are not evident in MR because it only has two
opposing labels without any explicit label correlation. However, our method takes
advantage of label correlations, which provide limited help on the MR dataset. Although

**Table 4 Performance of backbones with and without DALS.**

| Model | 20NG | | R8 | | R52 | | Ohsumed | | MR | |
|---|---|---|---|---|---|---|---|---|---|---|
| | Acc | Macro-F1 | Acc | Macro-F1 | Acc | Macro-F1 | Acc | Macro-F1 | Acc | Macro-F1 |
| TextCNN | 0.8502 (+0.0287) | 0.8465 | 0.9628 (+0.0057) | 0.9161 | 0.9166 (+0.0407) | 0.6841 | 0.6233 (+0.0389) | 0.5570 | 0.7917 (+0.0142) | 0.7917 |
| LSTM | 0.8143 (+0.0600) | 0.8123 | 0.9743 (+0.0134) | 0.9348 | 0.9441 (+0.0393) | 0.7173 | 0.6320 (+0.1210) | 0.5222 | 0.7789 (+0.0056) | 0.7799 |
| FastText | 0.8519 (+0.0581) | 0.8479 | 0.9743 (+0.0130) | 0.9302 | 0.9441 (+0.0160) | 0.7470 | 0.6588 (+0.0818) | 0.5874 | 0.7766 (+0.0252) | 0.7768 |
| TextGCN | 0.8634 (+0.0000) | 0.8585 | 0.9710 (+0.0003) | 0.9330 | 0.9381 (+0.0025) | 0.6808 | 0.6875 (+0.0039) | 0.6281 | 0.7608 (−0.0066) | 0.7608 |
| BertGCN | **0.8947 (+0.0017)** | **0.8888** | **0.9828 (+0.0018)** | **0.9487** | **0.9667 (+0.0007)** | **0.8419** | **0.7361 (+0.0081)** | **0.6603** | **0.8646 (+0.0046)** | **0.8647** |

**Note:**
Each bold entry denotes the best performance of the metric in the column.

**Table 5 Test accuracy and calibration performance of BertGCN with different loss functions.**

| Methods | R8 | | R52 | | Ohsumed | | MR | |
|---|---|---|---|---|---|---|---|---|
| | Acc | ECE | Acc | ECE | Acc | ECE | Acc | ECE |
| CE | 0.9810 | 0.012991 | 0.966 | 0.033264 | 0.728 | 0.244315 | 0.8600 | 0.105862 |
| LS | 0.9790 | 0.069798 | 0.9638 | 0.074365 | 0.7316 | **0.168407** | 0.8613 | 0.077004 |
| FL (*Lin et al., 2017*) | 0.9804 | 0.008995 | 0.9533 | 0.031282 | 0.6960 | 0.249241 | 0.8576 | 0.113956 |
| MbLS (*Liu et al., 2022*) | 0.9758 | 0.013138 | 0.9603 | 0.031092 | 0.7062 | 0.210890 | 0.8571 | 0.115159 |
| DALS | **0.9828** | **0.008458** | **0.9667** | **0.030494** | **0.7361** | 0.168829 | **0.8646** | **0.075474** |

**Note:**
Best results are highlighted in bold style.

some results cannot meet the desired performance on the MR dataset, the overall results remain competitive, which proves the effectiveness and flexibility of DALS.

## Accuracy and calibration performance with different losses

In this study, we also explored the test performance and calibration ability of DALS compared with other methods. The expected calibration error (ECE) (*Naeini, Cooper & Hauskrecht, 2015*; *Guo et al., 2017*) is a commonly used method for measuring calibration. The samples were evenly distributed in $M$ bins. $B_m$ represents the set of predicted samples belonging to the $m$-th bin. The average accuracy of the samples in $B_m$ is denoted as $Acc_m$, and the average confidence within $B_m$ is denoted as $Conf_m$. Here, we set $M = 10$.

$$\text{ECE} = \sum_{m}^{M} \frac{|B_m|}{N} |Acc_m - Conf_m|. \tag{9}$$

As shown in Table 5, the accuracy and ECE of the different methods are reported on the four datasets, and only the state-of-the-art BertGCN model is chosen for comparison. Our method achieves a higher accuracy than other existing methods. The performance gains suggest that DALS helps improve text classification models, such as BertGCN. The ECE results also show that our method achieves a lower ECE than most other methods, including CE, and enables the calibration of neural models. Although our ECE result on

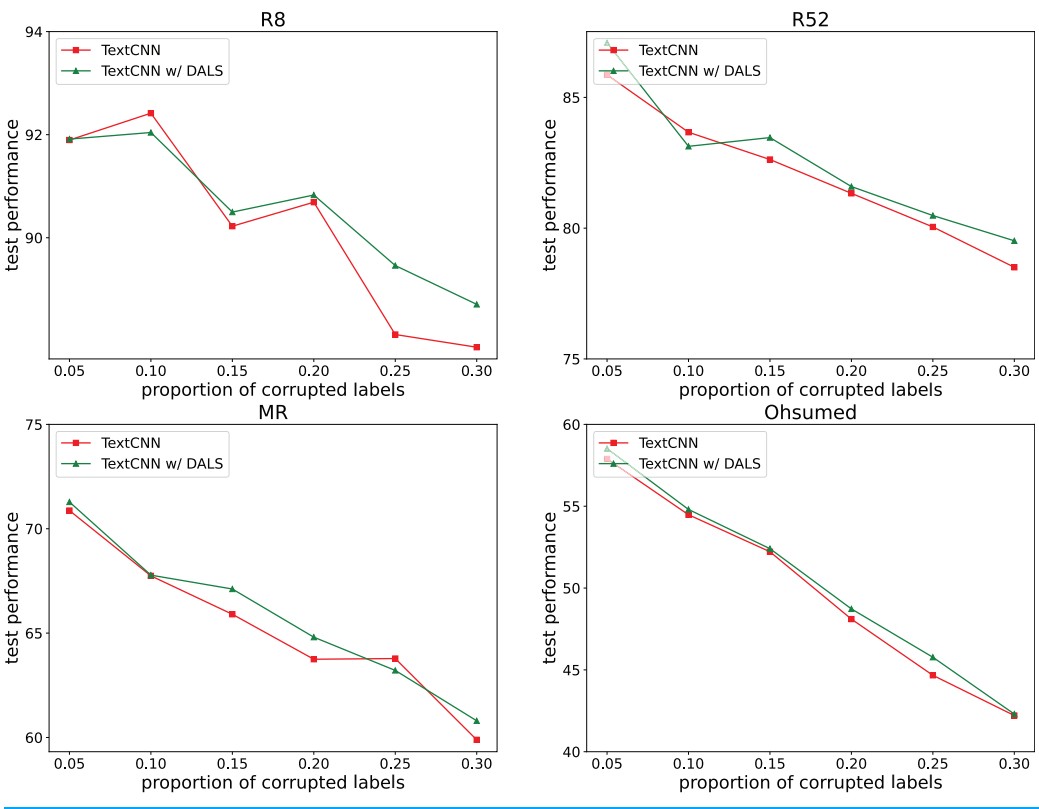

**Figure 2 Performance with different proportions of corrupted labels (TextCNN).**

Ohsumed is less satisfactory than that on LS, a balanced performance is achieved with higher accuracy.

## Effect of corrupted labels

To explore the ability of the model to deal with mislabeling, 5%, 10%, 15%, 20%, 25%, and 30% of the training data were randomly selected, and the labels were randomly replaced from among the remaining labels with the same transition probability. The test set remained unchanged. Figures 2 and 3 both show the effects of different proportions of corrupted labels on the test results. Figure 2 presents the results of the experiments conducted on TextCNN, whereas Fig. 3 uses TextGCN. In general, as the percentage of fake labels increases, the accuracy decreases. After using DALS, the performance of the backbone improved in most cases. These experiments prove that the proposed method maintains its robustness and reduces the negative impact of labeling errors on the model.

   Confusion matrices were generated on the Ohsumed dataset (Fig. 4), where each case tends to be classified into the C23 category of the backbone; therefore, the color of this column is darker. After applying DALS, the model reduces the predicted probability of C23, and the color of column C23 is lighter. This suggests that if the sample is often

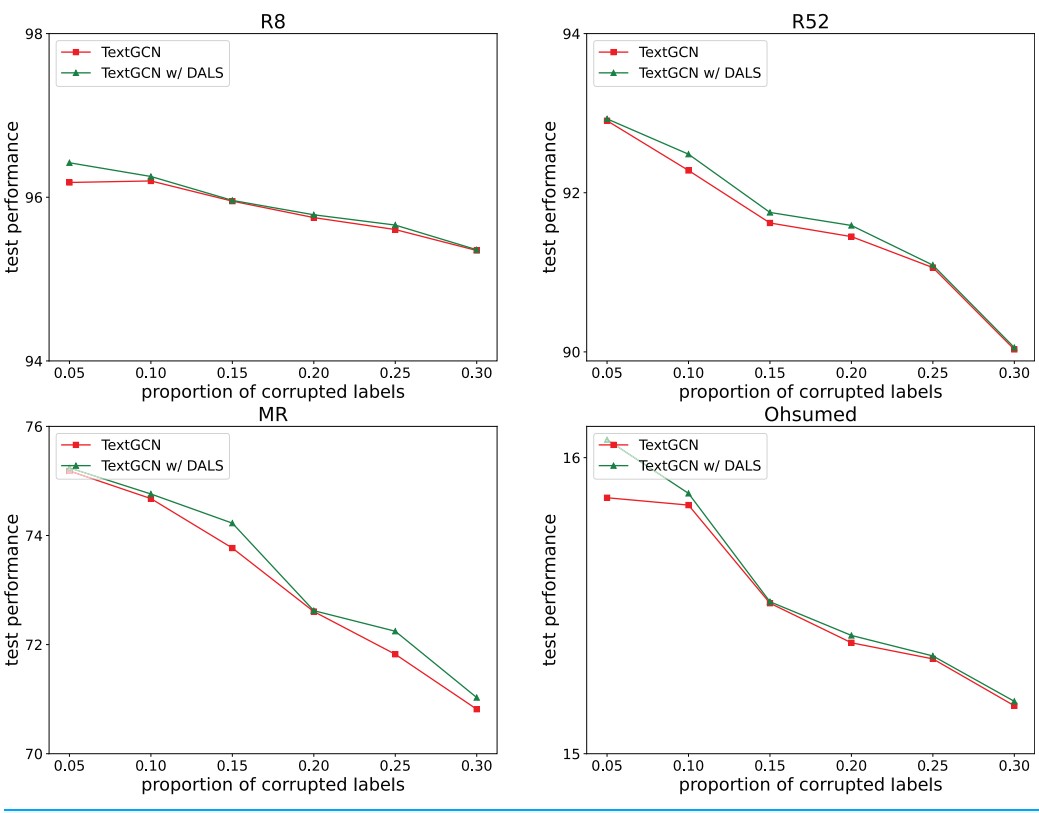

**Figure 3 Performance with different proportions of corrupted labels (TextGCN).**

misclassified into certain categories that are confusing, more supervision is required for that class instead of excessive regularization.

## DISCUSSION

### Effect of hyperparameters

Among the hyperparameters, $\alpha$ measures the contribution of soft and hard targets, which represents the degree to which model prediction deviates from the hard labels. As shown in Fig. 5A, when $\alpha = 0.96$, TextGCN with DALS achieves the best result on the Ohsumed dataset. When the value exceeds 0.96, the model performance degrades, caused by the small proportion of soft labels. The increase in the non-target distribution is too small to show a difference between labels. When $\alpha$ is lower than 0.96, the contribution of the soft label is higher, increasing error tolerance. This can easily cause underfitting, reducing the learning ability of the model. Figure 5A also shows that the highest accuracy of BertGCN with DALS is obtained when $\alpha = 0.95$. We also explored the settings of $\lambda$ to balance the impact of true positive and false positive cases, as shown in Fig. 5B. A $\lambda$ value of approximately 1.4 emerges as the optimal balance point, yielding the highest accuracy on the test set with the TextGCN method. Deviations towards smaller or larger values of $\lambda$ result in a decline in model efficacy. Tuning $\lambda$ to an appropriate value can effectively control the smoothness of the model, thereby enhancing the overall model by managing true positive and false positive instances.

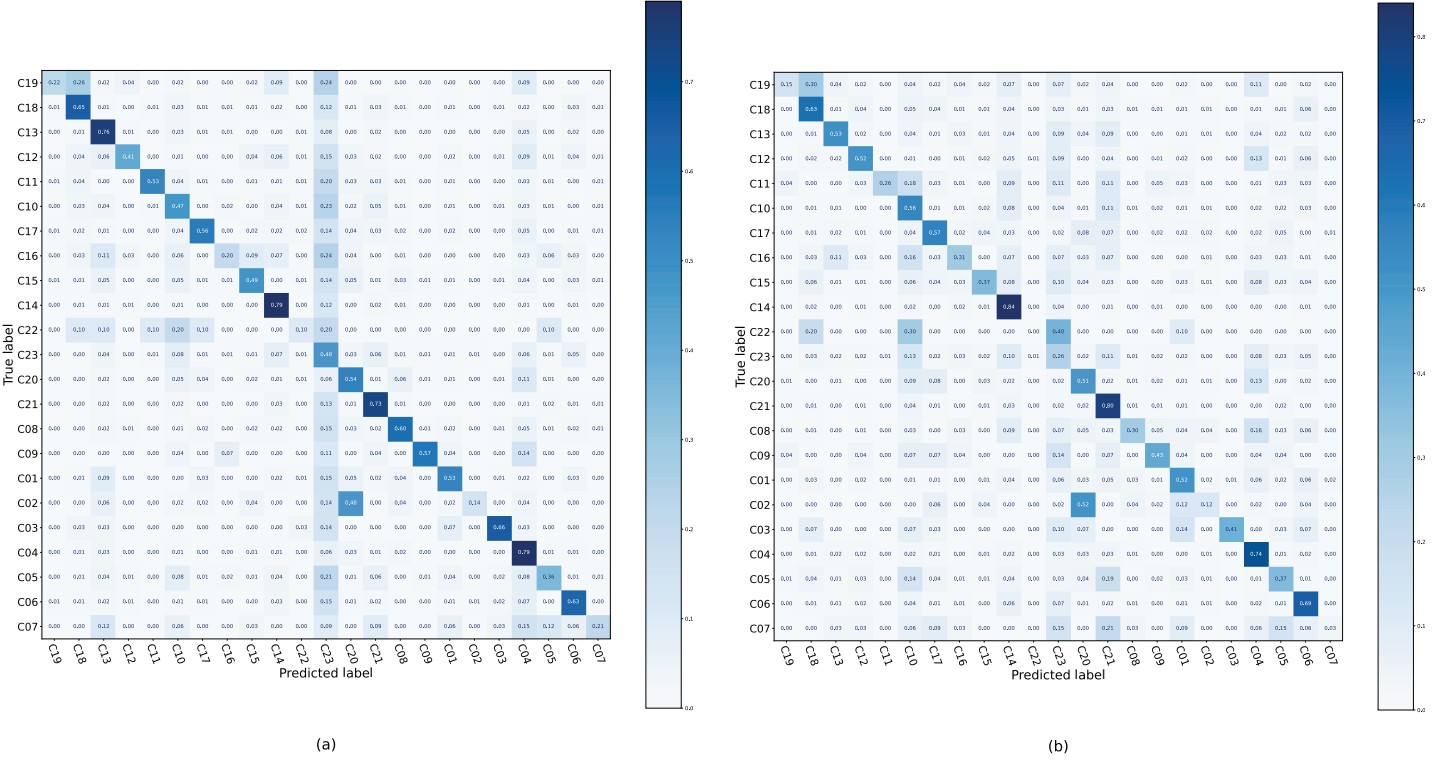

**Figure 4 Confusion matrix of TextCNN and TextCNN with DALS on the Ohsumed dataset.**

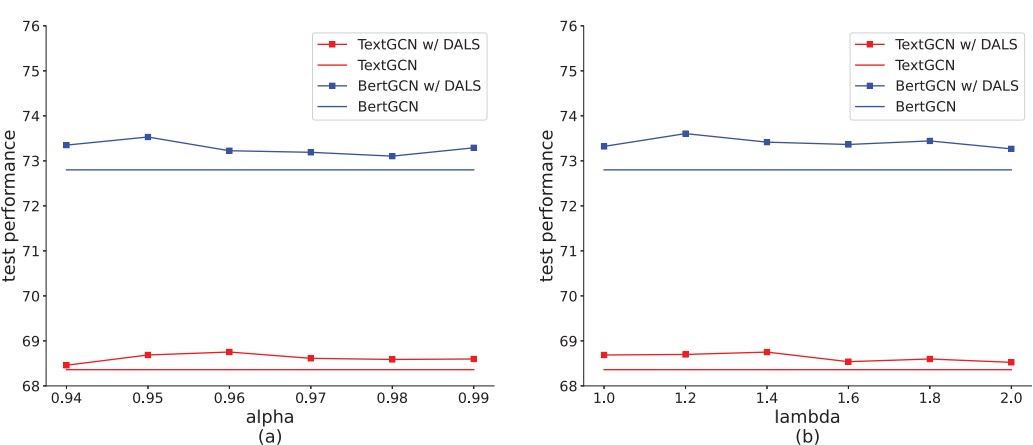

**Figure 5 Effect of hyperparameters on the Ohsumed dataset (TextGCN and BertGCN).**

## Connection with model complexity

We also investigated whether our regularization method affects the complexity of the model. The TextCNN model requires several kernels to capture different text features, with a default kernel size of 100. Our experiments reduced the number of neurons by reducing the number of kernels to 2, 4, 6, 8, and 10. The results shown in Fig. 6 indicate that the

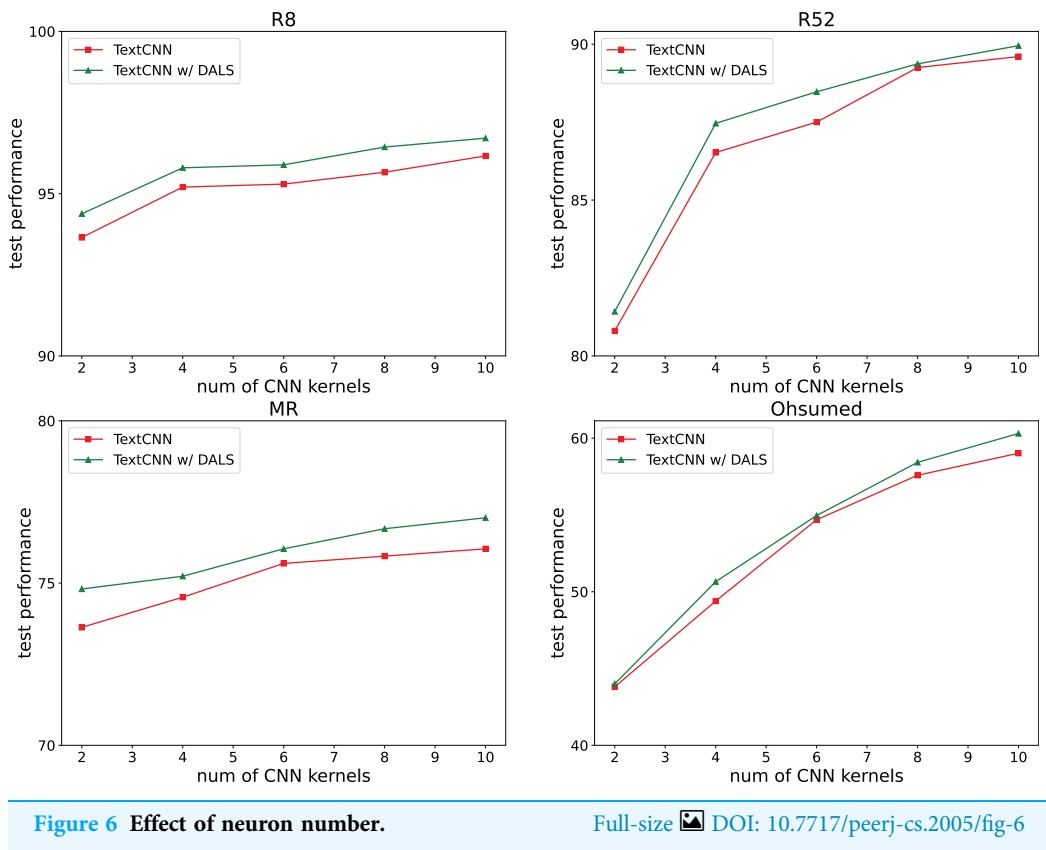

**Figure 6  Effect of neuron number.**

model captures more features with an increasing number of kernels, and its accuracy remains higher than that of the backbone.

## CONCLUSIONS

LS helps alleviate the problem of overconfidence and enhances the calibration ability of models. DALS, an adaptive LS method, offers a reasonable approach for obtaining the soft distribution of classes by employing true and false positive samples to iteratively learn their distribution scores. Experiments on five datasets show that DALS promotes classification performance, calibration ability, and model robustness. In summary, the advantages of our model are: 1) employing both true positive and false positive cases in learning smoothing parameters, thereby expanding the training data for model calibration and improving performance; 2) providing a LS approach *via* plug-and-play without any changes to the original models.

Our approach has several limitations that need to be considered for improvement: 1) DALS may not yield significant performance improvements for classification tasks with sparse data. In such cases, the model should focus more on data fitting than generalization; 2) model hyperparameters are dataset-dependent. Consequently, hyperparameter settings become essential during the learning process to ensure optimal performance across different datasets.

In future work, we plan to extend our research by integrating this method into machine learning pipelines for various applications in downstream tasks to measure the correlation

between the decision thresholds of these tasks and the adaptive LS method and improve performance.

### Funding

This work is supported by the Major Project of Philosophy and Social Sciences of the Ministry of Education (Grant No. 21JDA050), the Research Fund of National Language Commission (Grant No.YB145-2), the Guangdong Education Department Project Foundation (Grant Nos. 2017KTSCX064, 2023WTSCX017), the Guangdong Philosophy and Social Sciences Foundation (Grant Nos. GD20XZY01, GD24CWY11), the Guangdong University of Foreign Studies Project Foundation (Grant Nos. LAI202305, LEC2019ZBKT002, LEC2022ZBKT005), the Guangzhou Science and Technology Project Foundation (Grant No. 202201010717), the National Natural Science Foundation of China (Grant No. 61977032) and the Hainan Natural Science Foundation (Grant Nos. 620QN282, 621MS054). The funders had no role in study design, data collection and analysis, decision to publish, or preparation of the manuscript.

### Grant Disclosures

The following grant information was disclosed by the authors:
Philosophy and Social Sciences of the Ministry of Education: Grant No. 21JDA050.
Research Fund of National Language Commission: Grant No.YB145-2.
Guangdong Education Department Project Foundation: Grant Nos. 2017KTSCX064, 2023WTSCX017.
Guangdong Philosophy and Social Sciences Foundation: Grant Nos. GD20XZY01, GD24CWY11.
Guangdong University of Foreign Studies Project Foundation: Grant Nos. LAI202305, LEC2019ZBKT002, LEC2022ZBKT005.
Guangzhou Science and Technology Project Foundation: Grant No. 202201010717.
National Natural Science Foundation of China: Grant No. 61977032.
Hainan Natural Science Foundation: Grant Nos. 620QN282, 621MS054.

### Competing Interests

The authors declare that they have no competing interests.

### Author Contributions

- Han Ren conceived and designed the experiments, performed the experiments, analyzed the data, performed the computation work, prepared figures and/or tables, authored or reviewed drafts of the article, and approved the final draft.
- Yajie Zhao conceived and designed the experiments, performed the experiments, analyzed the data, performed the computation work, prepared figures and/or tables, authored or reviewed drafts of the article, and approved the final draft.
- Yong Zhang analyzed the data, authored or reviewed drafts of the article, and approved the final draft.

- Wei Sun conceived and designed the experiments, analyzed the data, authored or reviewed drafts of the article, and approved the final draft.

## Data Availability

The 20ng dataset is available at Zenodo: N/A. (2021). 20 news group (20ng) (Version v1) [Data set]. Zenodo. https://doi.org/10.5281/zenodo.7555237.

The R8 and R52 are most widely used data collections for text categorization. They are publicly available as part of the Reuters *Corpus* through Reuters, Inc. at GitHub: https://github.com/yao8839836/text_gcn/tree/master/data.

The Ohsumed dataset is available at Zenodo: N/A. (2021). OHSUMED (Version v1) [Data set]. Zenodo. https://doi.org/10.5281/zenodo.7555276.

The MR dataset is available at Zenodo: N/A. (2021). Movie review (MR) (Version v1) [Data set]. Zenodo. https://doi.org/10.5281/zenodo.7555273.

The GloVe embedding is available at Zenodo: Liebl Bernhard. (2021). GloVe 6B Vectors (1.2) [Data set]. Zenodo. https://doi.org/10.5281/zenodo.4925376.

## Supplemental Information

Supplemental information for this article can be found online at http://dx.doi.org/10.7717/peerj-cs.2005#supplemental-information.

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
