# Peer review of "Learning label smoothing for text classification"

_PeerJ Computer Science, doi:10.7717/peerj-cs.2005_

## Round 0.1 · original submission · Major Revisions

Dear authors,

Thank you for submitting your article. Reviewers have now commented on your article and suggest major revisions. When submitting the revised version of your article, it will be better to address the following:

1- The research gaps and contributions should be clearly summarized in the introduction section. Please evaluate how your study is different from others in the related section.

2- Please include future research directions.

3- The values for the parameters of the algorithms selected for comparison are not given.

4- The paper lacks the running environment, including software and hardware. The analysis and configurations of experiments should be presented in detail for reproducibility. It is convenient for other researchers to redo your experiments and this makes your work easy acceptance. A table with parameter settings for experimental results and analysis should be included in order to clearly describe them.

5- Please clarify the pros and cons of the methods. What are the limitation(s) methodology(ies) adopted in this work? Please indicate practical advantages, and discuss research limitations.

6- Minor English grammar and writing style errors should be corrected.

7- Explanation of the equations should be checked. All variables should be written in italic as in the equations. Equations should be used with correct equation numbers within the text.

8- Some more recommendations and conclusions should be discussed about the paper considering the experimental results. The conclusion section needs significant revisions. It should briefly describe the findings of the study and some more directions for further research. The authors should describe academic implications, major findings, shortcomings, and directions for future research in the conclusion section. The conclusion in its current for is confused in general. It is strongly suggested to include future research of this manuscript. What will be happen next? What we supposed to expect from the future papers? So rewrite it and consider the following comments:
- Highlight your analysis and reflect only the important points for the whole paper.
- Mention the benefits
- Mention the implication in the last of this section.

**Language Note:** The Academic Editor has identified that the English language must be improved. PeerJ can provide language editing services - please contact us at copyediting@peerj.com for pricing (be sure to provide your manuscript number and title). Alternatively, you should make your own arrangements to improve the language quality and provide details in your response letter. – PeerJ Staff

·

Basic reporting

The writing should be improved. For example, "Let D denote" must be "Let D denotes", "in the following" should be "Following it, ", "label smoothing introduces regularizer H(u,p)" should be "a regularizer H(u,p) is introduced in the smoothing process".

Experimental design

1.The model deals with positive cases and negative ones separately. What is the reason of doing so?
2.According to formula 9, the optimization objective is iteratively adjusted. When will the process stop?
3.The performances do not always increase when adding DALS to TextGCN, according to Table 4. Please give details about it.

Validity of the findings

no comment

Additional comments

Text classification is a traditional area of natural language processing. This paper aims to the problem of overconfidence in it and proposes a learning-based smoothing method in order to make the model more robust.
This paper is clear and easy to understand.

Cite this review as

·

Basic reporting

1. Most of the article is easy to read. However starting from line 193 something is a little blurry. For example, in Line 193, "ÿ+0 as uniform distribution and ÿ-0=0". But, in Line 176, "At the beginning of epoch t, ÿ+t and ÿ-t is initialized to a zero matrix." Which one is correct?
2. Please clearly define ÿ+0, d ÿ-0 and q0 in Line 194.
3. The literatures are well referenced and relevant. However, the information for some references is incomplete. For example, references in Lines 378 and 458 are incomplete.
4. At the end of each epoch, the proposed method average the accumulated class distribution row by row to balance the score contribution of each sample, and the use the formuls in Line 188 to calculate qck. Because |D1| and |D2| are different (maybe |D1| is a large number and |D2| is just one), using average is not ideal.

Experimental design

1. This research used accuracy to evaluate the performance of the classification models. However, in most of the related researches micro-f1 and macro-f1 are more suitable.
2. Please using 10-fold cross-validation in the experimental design. This evaluation method is relatively fair

Validity of the findings

1. This study achieved an average improvement of 5%. However, the author should describe the reasons for many unsatisfactory results?
2. All underlying data have been provided.

Additional comments

None

Cite this review as

---

## Round 0.2 · Minor Revisions

Dear authors,

Thank you for the revision. The reviewers did not respond and I checked the revision and response letter myself before making the decision. It seems that some minor edits are required for the quality of the paper.

1- Explanation of the equations should be checked. All variables should be written in italic as in the equations. Equations should be used with correct equation numbers within the text.
2- The space character should be used correctly.
3- Even if there is no comparison, the micro-f1 and macro-f1 scores of your proposed method in the data are expected to be given.

Best wishes,

---

## Round 0.3 · accepted · Accept

Dear authors,

Thank you for clearly addressing all the reviewers' and editor's comments. I confirm that the quality of your paper is improved. The paper is now ready for publication in light of the last revision.

Best wishes,